# Docking and Electronic Structure of Rutin, Myricetin, and Baicalein Targeting 3CLpro

**DOI:** 10.3390/ijms242015113

**Published:** 2023-10-12

**Authors:** Sergio A. de S. Farias, Kelvyn M. L. Rocha, Érica C. M. Nascimento, Rafael do C. C. de Jesus, Paulo R. Neres, João B. L. Martins

**Affiliations:** 1Laboratory of Computational Simulations (LabIn02), Institute of Educational Sciences, Federal University of Western Pará, Santarém 68040-255, Pará, Brazil; sergio.farias@ufopa.edu.br (S.A.d.S.F.); paulo.neres@ufopa.edu.br (P.R.N.); 2Instituto de Química, Universidade de Brasília, Brasília 70910-900, Distrito Federal, Brazil; kelvynmagalhaeslr@gmail.com (K.M.L.R.); ericacristinamoreno@gmail.com (É.C.M.N.); rafaelccampos30@gmail.com (R.d.C.C.d.J.)

**Keywords:** 3CLpro protease, SARS-CoV-2, rutin, myricetin, baicalein, density functional theory, docking

## Abstract

Understanding the role of 3CLpro protease for SARS-CoV-2 replication and knowing the potential of flavonoid molecules like rutin, myricetin, and baicalein against 3CLpro justify an investigation into their inhibition. This study investigates possible bonds and reactivity descriptors of rutin, myricetin, and baicalein through conformational and electronic properties. Density functional theory was used to determine possible interactions. Analyses were carried out through the molecular electrostatic potential, electron localization function, Fukui function descriptors based on frontier orbitals, and non-covalent interactions. A docking study was performed using a resolution of 1.55 Å for 3CLpro to analyze the interactions of rutin, myricetin, and baicalein. Scores of structures showed that rutin is the best ligand, followed by myricetin and baicalein. Docking studies showed that baicalein and rutin can establish effective interactions with residues of the catalytic dyad (Cys145 and His41), but just rutin forms a hydrogen bond. Myricetin, in turn, could not establish an effective interaction with Cys145. Baicalein interaction arose with active residues such as Arg188, Val186, Gln189, and Gln192. Interactions of rutin and myricetin with Arg188 and Gln189 were also found. A critical interaction was observed only for rutin with the hydroxyls of ring A with His41, and also for Cys145 with rings B and C, which is probably related to the highest score of rutin.

## 1. Introduction

3-Chymotrypsin-like protease (3CLpro) is highly preserved in the coronavirus family and is the central enzymatic machinery responsible for the viral replication cycle [1,2]. 3CLpro is responsible for the cleavage of viral protein with positively charged histidine residues (His41, His163, His164, and His172) [3]. Natural plant polyphenol molecules with low side effects represent an exciting option for treating patients affected by the virus [4,5,6,7,8].

In the last few years, different flavonoid studies have been applied to address the ability to inhibit SARS-CoV-2 3CLpro protein, relating the importance of catalytic dyad His41 and Cys145 [2,3,9,10,11,12]. Among these, rutin, myrecetin, and baicalein have been extensively studied for this target (Figure 1) [1,2,3,4,5,6,7,9,10,11,13,14,15,16,17,18].

Rutin (quercetin-3-rhamnoglucoside) is composed of quercetin (flavonol) with the disaccharide rutinose [8,13,14], and binding to 3CLpro can modify the spectroscopic properties and tertiary structure of protease [2]. Rutin sugar groups are structural and functionally relevant because of their hydrophilic nature [2]. The quercetin portion of rutin shows a fundamental importance in its binding interaction with 3CLpro [2]. Therefore, the flavonoid class of molecules deserves attention as antivirals and can be a promising starting point in developing and designing new antiviral drugs. However, it was recently evaluated whether a more hydrophilic chemical adduct allows better interaction with 3CLpro despite molecular weight increase [2,9]. It was suggested that rutin interacts indirectly with protease through quercetin rings A and C through residues His41 and Cys145, while ring B could perform hydrogen binding with Leu141 residue or with a chain of Glu166 residue [2].

Rehman and co-workers showed that rutin towards 3CLpro has a binding energy of −9.4 kcal/mol, forming one π-sulfur bond with Cys145 of 5.13 Å and conventional hydrogen bonding with His163 [15]. The 6LU7 structure was solved to a resolution of 2.16 Å [15]. Deetanya et al. studied rutin and baicalein and found the results inconsistent with previous in silico studies with a binding energy of −10.0 kcal/mol for rutin [3]. They predicted a binding mode of rutin against PDB:6LU7 with hydrogen bonding Lys141, Ser144, and Gly302 residues. The pair interaction energy of fragment molecular orbital (FMO) showed that Glu166 has the highest energetic contribution, followed by Glu166 > Cys145 > Lys141 > Val303 > His41 [3]. Rutin has shown a binding energy of −9.2 kcal/mol while myricetin has given a value of −7.4 kcal/mol using PDB:6LU7 [11]. Researchers found a π-sulfur of 5.03 Å, followed by a cation–π interaction of 4.06 Å, while conventional hydrogen bonding with Leu141 gave 2.50 Å [11]. 

It was shown that myricetin inhibits SARS-CoV-2 through covalently binding to 3CLpro [9], where the lateral chain of His41 is close to Cys145, acting as a nucleophile to attract hydrogens [9]. These suggestions were based on a PDB 7DPP of 2.10 Å for resolution with the ligand myricetin. Otherwise, the pyrogallol group of myricetin is supposed to work as an electrophile, covalently binding to Cys145 of the catalytic dyad [9]. As a result, myricetin has good inhibitory activity against 3CLpro [9]. Analyses of a fluorescent probe of 8-anilinonaphthalene-1-sulfonate (ANS) show that baicalein binds with a 3CLpro active site and that rutin could form a stable complex through a His41 and Cys145 dyad [3]. The interaction with catalytic dyad His41 and Cys145 through hydrogen bonding makes the docking efficient to inhibit 3Clpro [16,17]. Myricetin was studied using molecular docking, and a binding energy of −7.4 kcal/mol was found using PDB:6LU7 [11].

The baicalein carbonyl group established a hydrogen bond with the main chain of Glu166, while the free phenyl ring was inserted into the S2 subsite by making hydrophobic interactions with multiple residues Gln189/Arg188/Met49/Cys44/His41. Notably, apart from the hydrophobic interactions, the catalytic Cys145 and His41 also formed S–π (at 3.3 Å from the sulfur to the centroid of the phenyl ring) and π–π interactions with the aromatic rings of baicalein, respectively [10]. 

Baicalein docking using PDB 6M2N with a resolution of 2.20 Å showed interactions with Cys145, His41, Glu166, Lys141, and Gly143 [18]. Baicalein is a non-covalent protease inhibitor forming H-bonding with a pyrogallol group with Leu141/Gly143 [9]. However, the His41 lateral chain constantly forms π–π interactions with baicalein and myricetin chromone regions [9]. Apparently, baicalein prevents the access of catalytic dyad substrates inside the active site [3,9], binding to Leu141 and Gly143, as well as the lateral chains of Ser144/His163 [9]. 

Several molecular modeling and computational chemistry techniques have been used to study the activity and relationships of flavonols against important targets [9,13,14,19]. In the context of these molecules, 3CLpro has been the target of relevance in the last few years [1,2,3,4,5,6,7,8,15]. However, studies of rutin, myricetin, and baicalein [15] have a gap in electronic structure descriptors, which are needed to evaluate new insights on the possible centers for interactions. Most recently, 3CLpro [20] PDB 7JR3 with a resolution of 1.55 Å was determined [20]. Therefore, it is important to verify the binding modes of these molecules against the target 7JR3 [20].

This work aims to achieve important descriptors for designing new potential entities through electronic structure and molecular descriptors and studying the interactions through covalent hydrogen bonding and nucleophilic and electrophilic interactions. Moreover, 7JR3 PDB was used for a docking study. We have used density functional theory (DFT) to obtain Fukui functions, bond critical points of Atom in Molecules Theory (AIM), frontier orbitals, non-covalent interactions, electron localization functions, and molecular electrostatic potential. 

## 2. Results and Discussion

### 2.1. Conformational Analysis

The electronic structure of these complex molecules has a significant component from the conformational structure used. In this case, it is important to analyze the planarity of ring C and the torsion angle between rings B and C. Therefore, we have carried out conformational analysis using CAM-B3LYP/def2TZV level and the implicit solvent integral equation formalism Polarizable Continuum Model (IEFPCM) method. Rutin is formed by one flavonol (quercetin) and two sugars, while myricetin and baicalein are flavonols. Figure 2 shows that the studied flavonols are planar in the lowest conformational energies. A recent DFT study also showed that some flavonoid dihedral angles (τ) have a planar configuration in the lowest conformational energy. The conformation of the hydroxyls binding to **C3** and **C5** carbons tends to form intramolecular bindings with ketone [21]. 

Appendix A shows calculated dihedral angles for the studied molecules compared with the literature. The calculated dihedral angles are τ (**O1**-**C2**-**C1′**-**C6′**), ω (**C9**-**O1**-**C2**-**C3**), ϕ (**C10**-**C4**-**C3**-**C2**), and, for rutin, δ (**C5″**-**C6″**-**O6″**-**C1**‴). We obtained some angles of these species with smaller values than those found in the literature [8,22]. For myricetin, the result for τ angle is in line with the literature for a planar structure [23,24,25,26,27].

### 2.2. Electronic Analysis

Figure 3 shows the representation of the electron localization function (ELF) isosurface of quercetin (rutin), rhamnoglucoside (rutin), myricetin, and baicalein. The quercetin (rutin) ELF map shows high electron density between **R1** and **R3** radicals of ring A, highly located around the hydrogen atoms (**H68** and **H65)**. Ring B shows electron density around the hydrogen atoms (**H67** and **H69**). The few differences between the ELF maps of the quercetin part of rutin and quercetin appeared mostly in **R1** and **R3** radicals [13].

The ELF isosurfaces between myricetin show high density on **R1** and **R3** radicals around hydrogen **H27** and high electron density over ring A in hydrogen **H24**. Ring B shows high electron density in hydrogens **H26** and **H25**. Baicalein shows high density in rings A, C, and B around hydrogens **H22**, **H21**, and **H27**, respectively.

The molecular electrostatic potential (MEP) surface accounts for interactions in the molecule and is helpful in analyzing reactivity descriptor behaviors in non-covalent interactions. Figure 4 shows the MEP surfaces in kcal/mol of rutin, myricetin, and baicalein. The molecules show a different pattern regarding charge distribution. Ring B has a negative distribution in rutin and a positive distribution in baicalein, while myricetin is less homogeneous. Hydrogens of rutin in ring A have a positive moiety, and baicalein and myricetin are different due to the hydroxyls. The electropositive regions are associated with **R3** and **R6** radicals. However, baicalein MEP is dominated by electropositive regions in ring B. Electronegative regions of baicalein occur in the ketone group, specifically in **R1** and **R3** [13]. 

The Highest Occupied Molecular Orbital (HOMO) and Lowest Unoccupied Molecular Orbital (LUMO) are widely used frontier orbitals to study global reactivity descriptors, covering energies and isosurfaces. Table 1 shows the HOMO orbital energies, gap HOMO-LUMO, electronegativity, chemical hardness, global softness (*S)*, and global electrophilicity (ω). From global descriptors, it is possible to suggest that (i) the highest $\varepsilon_{H}$ occurs in rutin, so rutin is the most reactive as a nucleophile, (ii) myricetin and baicalein have the lowest $\varepsilon_{L}$, so they are more reactive as electrophiles, and (iii) the lowest gap occurs in myricetin. The lowest electron transfer resistance (η) and the highest tendency of electron acceptor (ω) occur in myricetin. The highest electronegativities (χ) occur in myricetin and baicalein. 

The susceptibility of an organic molecule to electrophilic attacks is directly related to the capability to donate electrons, represented by the HOMO orbital. Fukui functions are depicted in Figure 5. Positively charged regions are related to the tendency to electrostatic attraction and negative regions with a tendency to electrostatic repulsion. All three molecules tend to receive electrophilic attacks in the ketone group and carbon **C2**. 

The capability to receive electrons is represented by the LUMO orbital, the electrophile, expressing suitability to accept a nucleophilic attack. Figure 5 shows that (i) carbon **C3** tends to accept nucleophilic attacks in rutin and myricetin; and (ii) baicalein tends to accept nucleophilic attacks in carbons **C5** and **C6** and also in the oxygens of radicals **R1** and **R2**. 

Analysis of energy density ($H_{c}$) and the electronic density distribution ($\rho_{c}$) following the work of Cremer and Kraka [28,29] was used to study the covalent versus electrostatic character in a weak bond [30]. Bond critical point (BCP) (3, −1) can be used to investigate intramolecular hydrogen bonding, which is important for the interaction of the molecule with the target [13,30]. If $H_{c}<0$, the bond is expected to have a component of covalent interaction. If $H_{c}>0$, the bond is non-covalent. The bond degree can be classified by means of the expression $BD=H_{c}/\rho_{c}$, the total energy by electron density in BCP [19,31,32,33,34]. Figure 6 and Table 2 show BCP (3, −1) and the values of $H_{c}$ and $H_{c}/\rho_{c}$ for the studied molecules. The critical points (3, −1) are identified as CP 1, 2, 3, and 4 (Figure 6).

The BCP (3, −1) formed between the hydrogen of the hydroxyl (**R1**) and the oxygen of the ketone gives a value of $H_{c}<0$ for myricetin and baicalein, but not rutin. Results from the literature show that quercetin has covalent interactions ($H_{c}<0$) [13]. However, rutin has no covalent interaction between **R1** and oxygen and showed BCP (3, −1) between **R5** and **R6**. The difference between these molecules is the rhamnoglucoside group. Therefore, the differences in BCPs (3,1) depend on rhamnoglucoside [13].

Myricetin and baicalein both give $H_{c}<0$ (a covalent component) between the hydrogen of the hydroxyl (**R2**) and ketone oxygen (**R1**). The results of $H_{c}/\rho_{c}$ show that the covalency degree of baicalein is higher than myricetin [13]. 

Weak interactions (non-covalent interactions) are mapped by the critical points (Figure 7). The blue region is a strong interaction, H–bond, and halogen–bond. The green interactions are van der Waals (vdW), and red is the steric effect. Three interactions in (3, −1) of rutin are localized between vdW and the steric effect, while the vdW interactions are between quercetin and rhamnoglucoside. The interaction between oxygen **O1** and hydrogen **H67** is important for the structure of these flavonoids. The highest differences between rutin and quercetin consist of repulsion forces (3, +1), or saddle points, between quercetin and rhamnoglucoside [13].

Non-covalent interactions (NCIs) are shown in Figure 7. Myricetin shows interactions relative to (3, −1) CP 1, 2, and 3 (Figure 6), as expected. However, in NCIs, interactions occur between radicals **R4** and **R5** and **R5** and **R6**. These interactions are also depicted in unprecedented form between oxygen **O1** and hydrogen **H26** in myricetin and baicalein. Baicalein interaction (3, −1) CP 1 was identified in Figure 6, but interactions not identified by critical points are found in Figure 7, between **R1** and **R2**, and between **R2** and **R3**.

### 2.3. Docking

Table 3 shows the docking scores and properties of studied flavonoids obtained by each molecule in their respective conformation.

Considering the data in Table 3, it is possible to note that the ligand capable of achieving the conformation with the best score is rutin. This ligand has highest molecular volume and size (Table 3), degree of freedom, and acceptor and donor groups of hydrogen bonding. The highest number of hydroxyls scattered for the entire molecular structure allows more effective interactions with the target protein.

The results of the other flavonoids reinforce the hypothesis that higher degrees of freedom and higher available hydrogen bonds are related to the best score obtained by rutin. The flavonoid with the second-best score is myricetin, which has the second-highest degree of freedom and the second-highest hydrogen bond interactions. However, the baicalein score obtained is the lowest with the lowest degree of freedom and sites of hydrogen bond available, according to Table 3. Baicalein IC_50_ from the experimental inhibitory activity is 0.39 ± 0.12 μM [18], while the docking score gives −8.28 kcal/mol for baicalein and −8.36 kcal/mol for myricetin using PDB 6LU7 [18]. Our results are in agreement with the IC_50_ of myricetin, which is 0.63 ± 0.01 μM, and is better than baicalein, with 0.94 μM [9]. 

The interactions established in a bi-dimensional representation are analyzed to understand better the scores obtained for each flavonoid (Figure 8). Baicalein can develop effective interactions with the residues of the catalytic dyad Cys145 and His41 (Figure 8). The interactions formed are π-sulfur for cysteine and π–π stacking (T form) and π-alkyl for histidine, in agreement with the literature results [3]. Due to the prominence of interactions involving π electrons, the aromatic center of baicalein presents itself as an important fragment to apply baicalein in 3CLpro. This center is common to all flavonoids since they have the same precursor, phenylalanine [35].

There are also expressive interactions of baicalein with different residues in the active site, like Arg188, Val186, Gln189, and Gln192. The interactions found are conventional hydrogen bonds and carbon–hydrogen. It is essential to note that the **R1** site where these interactions are depicted is also where the single BCP (3, −1) is located (Figure 6).

Additionally, the **R1** site concentrates on high electronic density, evidenced by the ELF isosurface (Figure 3), reinforcing its importance in intermolecular interactions performed by this ligand. Therefore, **R1** is a unique hydroxyl group that effectively forms conventional hydrogen bonds with Arg188, Gln192, and Val186 residues of baicalein, which could be explained by the affinity generated by its high negative charge density.

In addition, the Fukui functions suggest that the **R1** site is favorable to receive nucleophilic attacks (Figure 5). The **R2** site also shows favorability to the same type of attack and concentrates an elevated degree of negative charge density, as shown in Figure 5. So, the **R2** site presents the potential to form more weak interactions, having a lower priority than the **R1** site. The bi-dimensional representation presented in Figure 8 confirms that the **R2** radical is a unique radical that establishes a similar interaction to the hydrogen bond beyond the **R1** radical.

When the results of baicalein are compared, it is possible to identify one disagreement regarding the interaction with Cys145, which is with rings A and C, while in the literature, it is with ring B [18], probably due to the difference in the resolution of proteins. The 7JR3 protein used in the present study has better resolution (1.55 Å), which may relate to higher accuracy results. 

Figure 8 also shows the bi-dimensional docking conformation of quercetin flavonoid in comparison with rutin. As with baicalein, quercetin establishes a strong interaction with the Cys145 residue, but for quercetin, the nature of the interaction is hydrogen bonding. 

The Leu141 residue also establishes a strong short-range hydrogen bond interaction, showing considerable closeness with quercetin. Results in the literature suggest that this residue occupies a prominent position for inhibitory activity, aiding the effectiveness of the docking in accordance with the literature [18].

The correlation between the number of hydrogen bond interactions and the numbers of hydrogen acceptor and donor atoms established with the best docking score is reinforced. Even in the case of quercetin, this flavonoid does not interact effectively with the catalytic residue His41. Quercetin overcame (by a slight margin) the score obtained by baicalein once quercetin presented five hydrogen bonds (three of them with bond lengths under 2.50 Å). In contrast, baicalein performed three effective hydrogen bond interactions. However, comparing the other flavonoids studied may better shed light on this topic. 

Myricetin is less effective in establishing interactions with Cys145 because the interaction with the residue Met49 occupies the π-sulfur site. Furthermore, myricetin does not interact with the catalytic dyad since it demonstrates a higher attraction with the side chain of Met49. Myricetin shows effective short-range hydrogen bonds involving residues like Cys44 (2.77 Å), Val186 (2.58 Å), Arg188 (2.29, 2.49 Å), Gln189 (2.53 Å), and Gln192 (2.30 and 2.35 Å). These interactions appear to contribute favorably to the best score of myricetin compared with baicalein and quercetin.

The interaction with residue His41 of the catalytic dyad also occurs with the action of the aromatic center, as was confirmed in the 3CLpro–baicalein complex. This interaction reinforces the importance of aromaticity for the system to create effective interactions between the ligand and the catalytic dyad in the target protein. As all flavonoids presented aromatic centers, this opens the possibility of investigating more molecules of this class than those in this study.

### 2.4. Analyzing Electronic Structure against Docking Information

Furthermore, when comparing MEP isosurfaces, it is possible to notice the intense electronic density adjacent to the non-conjugated ring of baicalein and rutin, which enables interaction with the residue Cys145 (rutin) and His41 (myricetin). However, this interaction showed a weak nature at around 4.9 Å (Figure 9). This greater distance, plus the low-intensity interaction, contributed to the low effectiveness of myricetin in interacting with the catalytic dyad residue Cys145. On the other hand, myricetin shows a low intense electronic density of its non-conjugated ring, which can contribute to its removal from the Cys145 side chain. It is also possible that there exists a degree of competition for the interaction between His41 and Cys145, shown by the docking of myricetin (Figure 9).

Analyzing the ELF of myricetin (Figure 5), areas of more significant electron localization of **H24** and **H27** atoms are not involved in expressive interactions in the docking conformation. Therefore, this suggests that the molecular structure has smaller contributions to the interaction when applied to 3CLpro. 

In addition to the peculiar electronic densities of each molecule, the capability of ring B of myricetin to form a higher number of hydrogen bonds compared to baicalein, because of its higher number of hydroxyls, seems to be responsible for a slightly better score of myricetin, in agreement with the literature [9].

The Fukui functions of myricetin reveal that the favorable sites for nucleophilic and electrophilic attacks are concentrated around the ring center. These sites establish evident interactions with residues of the target protein. Furthermore, there was no clear relationship between interactions and myricetin BCP sites. Considering the results in Table 3, the conformation from the docking study of rutin was the most effective (Figure 7).

In a first analysis, it is possible to verify that rutin is the only ligand capable of forming a hydrogen bond with the catalytic dyad residue His41/Cys145. The best docking pose of these ligands shows the relevance of the active site (Figure 10). The better rutin score can be justified because of its interaction with higher intensity compared to the other interactions established for the other studied flavonoids. In addition to the interaction with His41, another interaction was identified with the residue His163, which is also positively charged [3] and possibly related to a more effective ligand positioning.

The quercetin part of rutin establishes interactions with the catalytic dyad in a form similar to myricetin. In contrast, the rhamnoglucoside moiety shows hydrogen bonds with other auxiliary residues (Arg188 and Gln189). The interaction with the residue Leu 141 is essential for the inhibitory effect [3].

The non-conjugated ring also concentrates a nucleophilic region according to the Fukui functions presented in Figure 5, which increases its affinity for residues with sulfur and, therefore, for the residue Cys145. Structurally, rutin seems to be the molecule with the higher capability to stabilize the system.

## 3. Materials and Methods

The molecules (Figure 1) were selected in the PubChem data bank: rutin (CID 5280805, C_27_H_30_O_16_), myricetin (CID 5281672, C_15_H_10_O_8_), and baicalein (CID 5281605, C_15_H_10_O_5_).

All electronic properties were calculated through density functional theory (DFT) with the Gaussian09 program package in the gas phase (vacuum) [36]. The CAM-B3LYP functional was used in conformational and electronic properties with Ahlrichs def2TZV as the basis set (CAM-B3LYP/def2TZV) [37,38].

Conformational analysis was carried out while varying the dihedral angle defined by atoms **O1**-**C2**-**C1′**-**C6′** with 18 steps of 10°. The study of electronic properties consisted of (i) non-covalent interactions (NCI), (ii) BCP, (iii) ELF, and (iv) MEP isosurface maps. All properties were performed using the AIMII package [39] and the Multiwfn package [40]. MEP was calculated using an electron density of *ρ* = 0.001 e/bohr^3^. The electrophilic and nucleophilic reactivities indices were calculated through Fukui functions using Multiwfn with densities of ρ=0.007 e/bohr^3^ for both $f_{N_{0}}^{-}$ and $f_{N_{0}}^{+}$.

An alternative methodology to study molecular reactivity descriptors involves using Fukui functions to describe local electronic densities [41]. The Fukui function that describes the removal of electrons from a molecule is $f^{-}$ (an electrophilic attack), and the Fukui function that describes the addition of electrons from a molecule is $f^{+}$ (a nucleophilic attack). These functions are mathematically described as follows: (1)$$f_{N_{0}}^{-}=\rho_{N_{0}}\left(\vec{r}\right)-\rho_{N_{0}-1}\left(\vec{r}\right)≅\rho^{HOMO},$$
(2)$$f_{N_{0}}^{+}=\rho_{N_{0}+1}\left(\vec{r}\right)-\rho_{N_{0}}\left(\vec{r}\right)≅\rho^{LUMO},$$
where ρ is the charge in $N_{0}-1$ cationic, $N_{0}+1\;$anionic, or $N_{0}$ neutral sites [42]. HOMO and LUMO molecular frontier orbitals are related to Fukui functions. Fukui functions have been used to predict electrophilic and nucleophilic sites [40]. Besides the local descriptors of Fukui functions, global descriptors like the electronic chemical potential (μ), chemical hardness (η), and electronegativity (χ) were analyzed. HOMO ($\varepsilon_{H}$) and LUMO ($\varepsilon_{L}$) orbital energies carry information about molecule reactivities. Higher $\varepsilon_{H}$ indicates strong activity in electrophilic reactions, while lower $\varepsilon_{L}$ indicates nucleophilic reactions [43].

Conformational analysis of dihedral angles between benzo-pyran (a heterocyclic compound formed by ring A and pyran ring C) and ring B of flavonols are fingerprints of intramolecular interactions. Topological analysis, in turn, provides critical points of weak bonds by means of a bond critical point (BCP) [32], making it possible to measure the degree of covalency in each BCP. The analysis of covalent bonds could be performed by the electron localization function (ELF) in contour maps [44]; an isosurface lower than 0.5 ELF is indicative of delocalized electrons, and regions with high electronic densities are indicative of hydrogen interactions or electron pairs [41,45]. The positive region in the molecular electrostatic potential (MEP) tends to interact with the negative region in the electrostatic surface molecular potential of other molecules. Thus, MEP tends to indicate nucleophilic and electrophilic active sites [46]. 

In order to evaluate the interactions between myricetin, baicalein, and rutin when applied to 3CLpro (7JR3 with a resolution of 1.55 Å) [20], the AutoDock Vina 1.1.2 algorithm was used [47] to elucidate the scores. The crystallographic structure of the protein was obtained from the Protein Data Bank, and the respective hydrogens were added using MolProbity, an online tool [48]. The suite used was the AutoDock 4 Tools package [49]. 

Calculations were performed assuming 1000 conformations with an exhaustiveness of 32, using a grid box with dimensions in *x*, *y*, and *z* axes of 72, 54, and 64, respectively. The center setup was 6.521, 23.315, and 23.939 in *x*, *y*, and *z* axes, respectively. This setup allowed the catalytic dyad His41/Cys145 to be in the center and included other relevant residues.

To perform the calculations, the protein was set as rigid while the ligand was set with all degrees of freedom unconstrained. This configuration led to finding a conformation with higher effectiveness in interactions between flavonoids and the target protein.

The conformation with the best score was chosen after the calculations, and the complexes generated were obtained by Discovery Studio 2017 [50].

## 4. Conclusions

We have studied three flavonoids through molecular docking and electronic structure calculation. For this study, we used the 7JR3 protein, which has better resolution (1.55 Å) than that presented in the literature. 

The present study shows that the catalytic dyad His41/Cys145 is present in baicalein and rutin interactions. The Cys145 residue binds via π-sulfur in both molecules, while the His41 residue binds by a carbon–hydrogen bond in baicalin and by a conventional hydrogen bond in rutin. Despite the catalytic dyad His41/Cys145 occurring in myricetin, this weak interaction is a π–π stacking for His41 and van der Waals for Cys145. 

Score conformation showed that rutin is the best ligand, followed by myricetin and baicalein. Rutin presents four critical points with *H_c_* > 0, with one single *H_c_*/*ρ*_c_ < 0. The descriptors that explain the best score conformation are: (i) accepting groups, (ii) hydrogen bond donors, and (iii) a higher number of hydroxyls throughout the molecular structure. The ELF isosurface, docking, and local descriptor analyses (Fukui functions) of baicalein suggest that the hydroxyl **R1** forms hydrogen bonding with Arg188, Gln192, and Val186 residues. The non-conjugated ring of rutin has an expressive electronic density. It also concentrates a nucleophilic region according to the Fukui functions, which increases its affinity for residues with sulfur and seems to favor interaction and affinity for the residue Cys145.

## Figures and Tables

**Figure 1 ijms-24-15113-f001:**
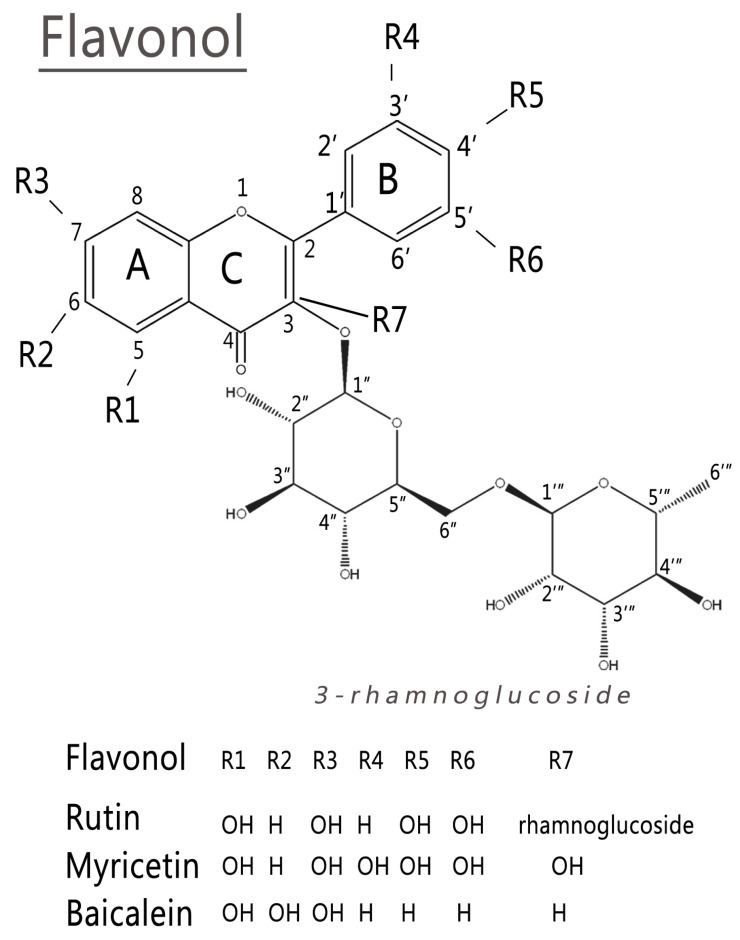
Structure of rutin, myricetin, and baicalein molecules.

**Figure 2 ijms-24-15113-f002:**
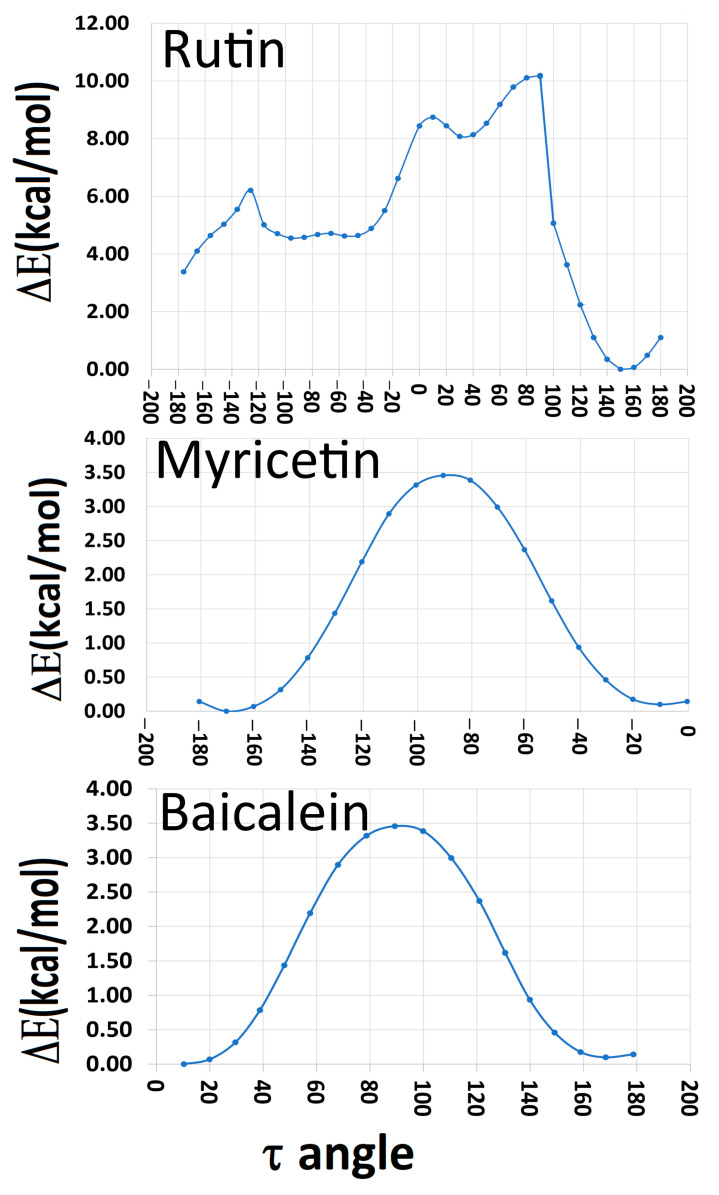
Conformational analysis using τ torsion angle of the investigated molecules (in degrees).

**Figure 3 ijms-24-15113-f003:**
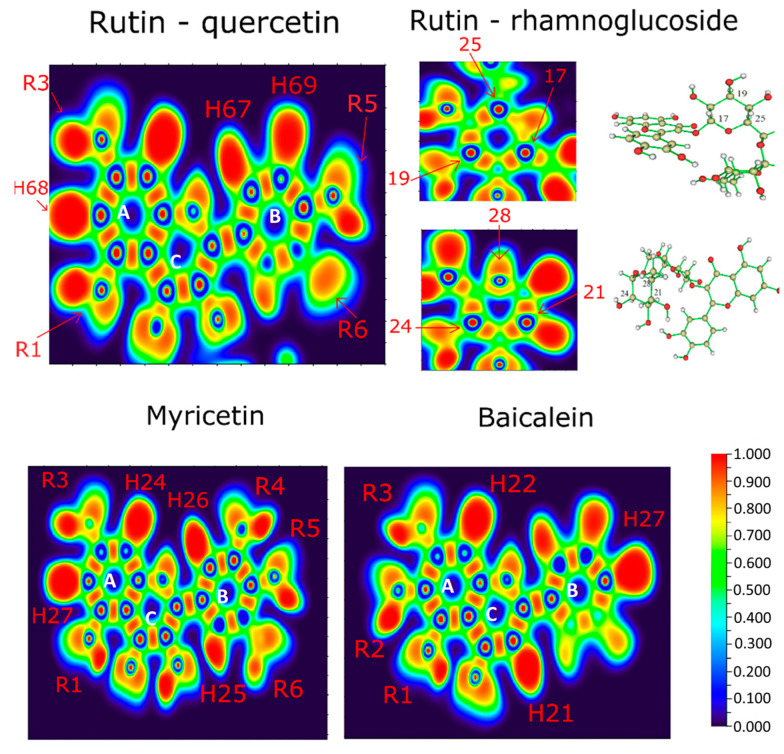
ELF contour maps of quercetin moiety of rutin, the rhamnoglucoside of rutin, myricetin, and baicalein.

**Figure 4 ijms-24-15113-f004:**
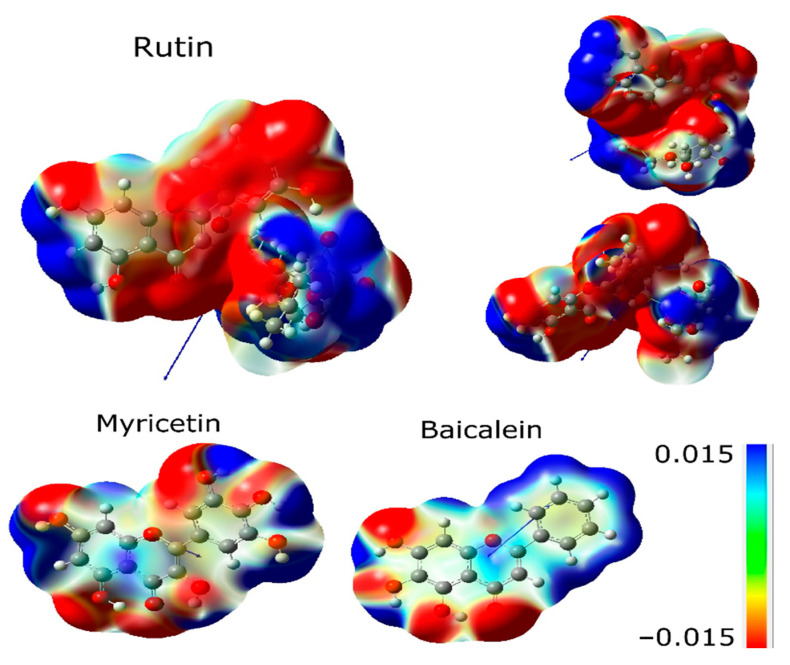
MEP surface of rutin, myricetin, and baicalein with scale.

**Figure 5 ijms-24-15113-f005:**
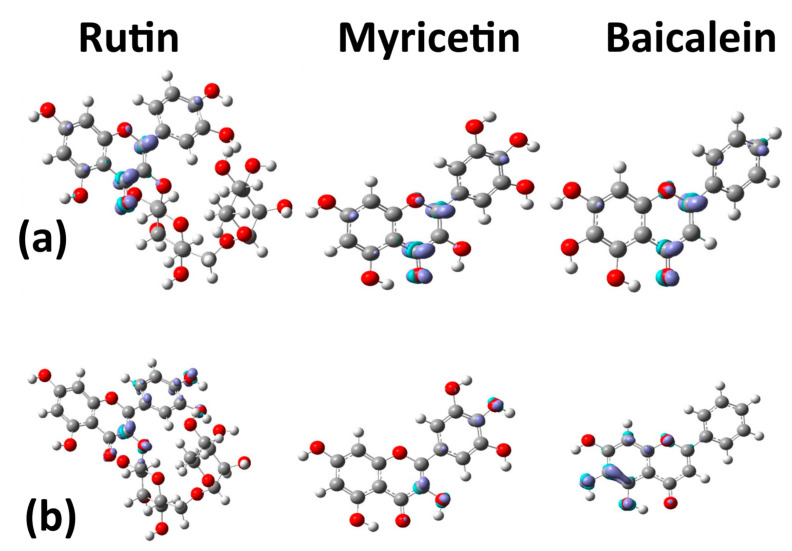
(**a**) Fukui functions $f^{+}$ of the three molecules to favor sites for electrophilic attacks. (**b**) Fukui functions $f^{-}$ of the three molecules to favor sites for nucleophilic attacks. Cyan is the positive and purple is the negative isosurface. Atom labels: red (oxygen), gray (carbon), and white (hydrogen).

**Figure 6 ijms-24-15113-f006:**
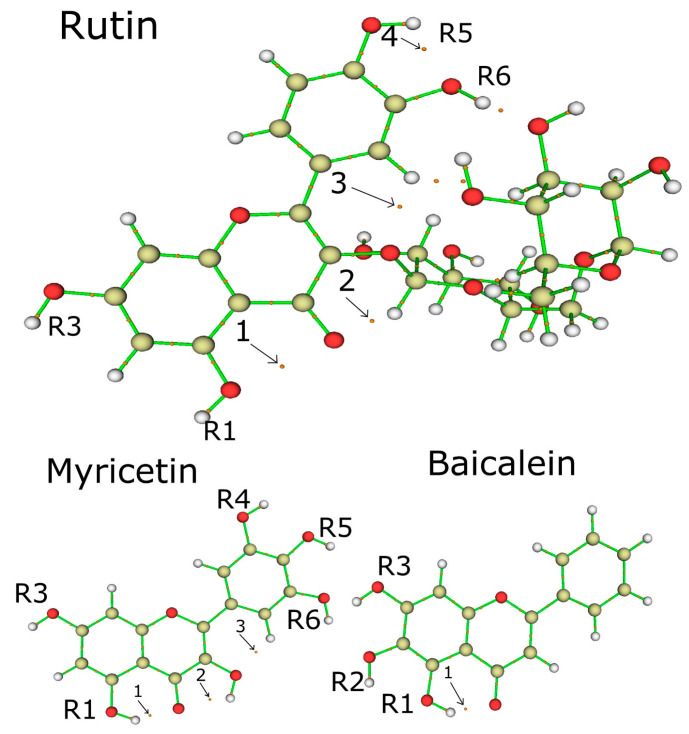
Bond critical point (3, −1) found for the three molecules.

**Figure 7 ijms-24-15113-f007:**
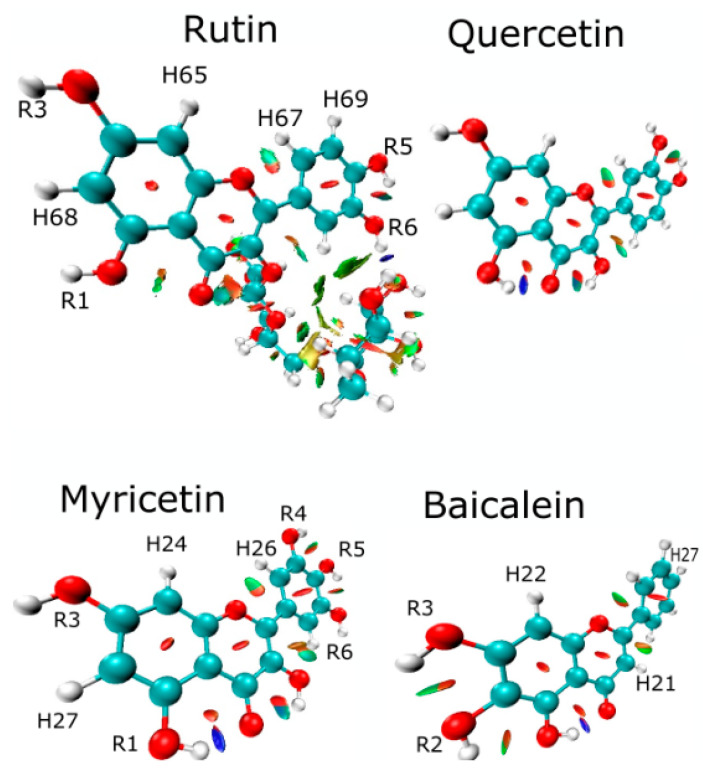
Non-covalent interactions in rutin, myricetin, and baicalein. Quercetin is shown for comparison.

**Figure 8 ijms-24-15113-f008:**
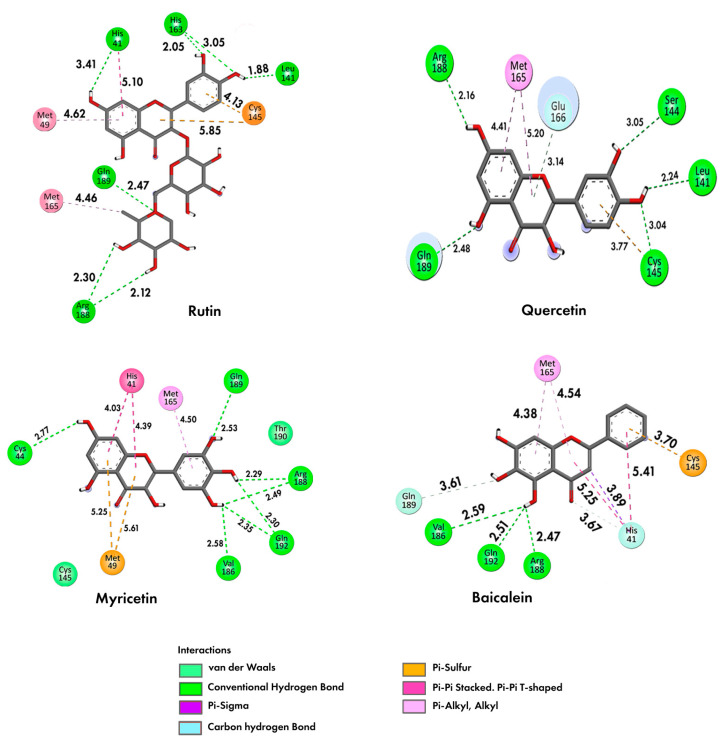
Bi-dimensional representation of the docking conformation of baicalein, quercetin, myricetin, and rutin.

**Figure 9 ijms-24-15113-f009:**
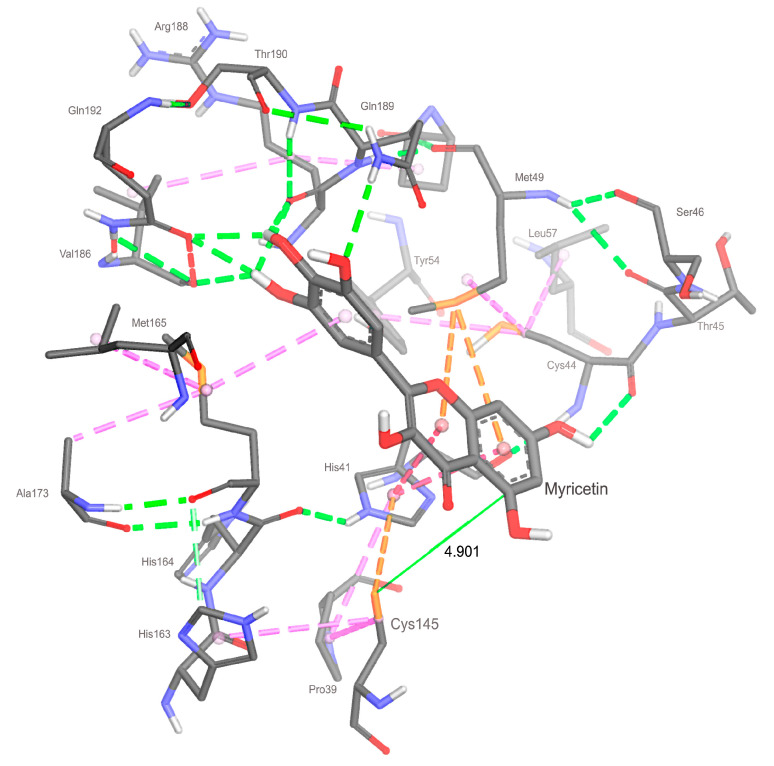
Tridimensional representation of the docking conformation of myricetin. The distance estimate for the van der Waals interaction with residue Cys145 is highlighted. The color labels are the same of Figure 8.

**Figure 10 ijms-24-15113-f010:**
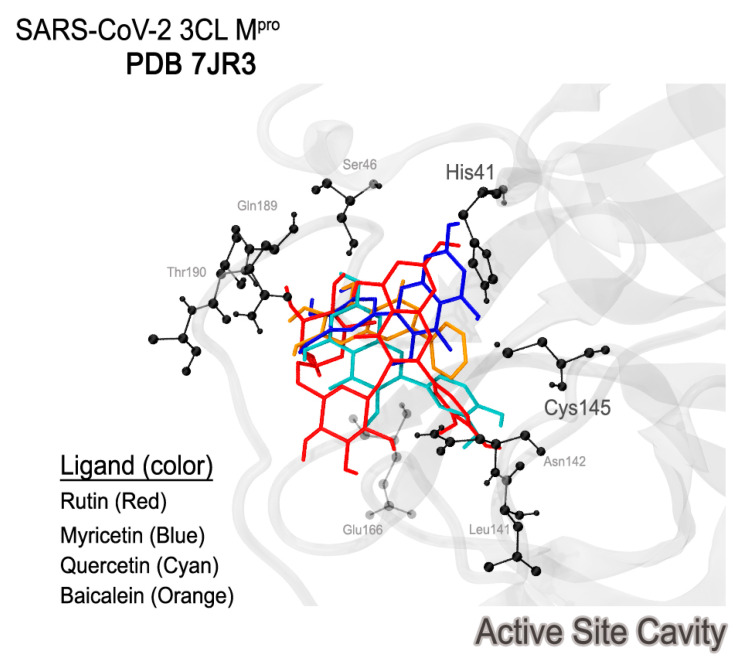
Best docking poses of the three ligands into the protein active site.

**Table 1 ijms-24-15113-t001:** Global reactivity descriptors of calculated molecules using CAM-B3LYP/def2TZV. Orbital energies in kcal/mol.

	$\varepsilon_{HOMO}$	$\varepsilon_{LUMO}$	$\Delta \varepsilon_{L-H}$	χ	η	*S*	ω
Rutin	−159.74	−15.54	144.20	87.64	72.10	0.001	53.27
Myricetin	−172.52	−33.82	138.70	103.16	69.35	0.001	76.73
Myricetin *	−171.51	−24.64	146.87	98.08	73.43	0.001	65.50
Baicalein	−179.15	−30.03	149.12	104.58	74.56	0.001	73.35

* Different from the first entry of myricetin, the second entry was calculated using the ligand structure deposited in RCSB:7DPP, without optimization.

**Table 2 ijms-24-15113-t002:** Topological properties of the electronic density in BCP (3, −1) of Figure 6, $H_{c}\left(a.u.\times 10^{-3}\right),$ and $H_{c}$/$\rho_{c}$ in parenthesis using CAM-B3LYP/def2TZV.

	1	2	3	4
Rutin	3.12 (0.24)	2.55 (0.13)	2.88 (−0.15)	3.57 (0.17)
Myricetin	−4.88 (−0.11)	2.96 (0.13)	3.31 (0.17)	-
Myricetin *	3.39 (0.22)	2.95 (0.13)	-	-
Baicalein	−6.86 (−1.47)	-	-	-

* Different from the first entry of myricetin, the second entry was calculated using the ligand structure deposited in RCSB:7DPP, without optimization.

**Table 3 ijms-24-15113-t003:** Results of scores obtained by docking calculations, structure properties, and the partition coefficient of studied flavonoids.

	Score (kcal/mol)	MW (g/mol)	Rotatable Bonds	Haceptor	Hdonor	Log *p*
Baicalein	−7.0	273.26	1	5	3	0.28
Quercetin	−7.1	302.24	1	7	5	1.23
Myricetin	−7.4	319.24	1	8	6	0.27
Rutin	−8.6	610.52	6	16	10	−1.29

## Data Availability

The datasets generated for this study can be found in the Appendix A.

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
