# Peer review of "Docking and Electronic Structure of Rutin, Myricetin, and Baicalein Targeting 3CLpro"

_ijms, 2023, doi:10.3390/ijms242015113_

Round 1
Author Response
Reviewer #1
- The term “reactivity” should be changed in the abstract and keywords because the authors investigate the binding modes of the three selected ligands but not their reactivity with the enzyme. The authors calculate reactivity descriptors on the different molecules, but they do not perform calculations of their reactivity.
Authors:
We kind acknowledge the referee. We have changed the sentences where the term reactivity appears.
- The scheme of the three ligands indicating rings A, B, and C would be appropriate before mentioning those rings in the text. In those schemes, the numbers used for atoms and molecular groups must be added from the beginning of the article. Otherwise, it is difficult to follow the description of the different molecular descriptors, although there is some numbering in some Figures. So, the Figure in the Materials and Methods section should be moved to the main text and be improved.
Authors:
The scheme of the three ligands indicating rings A, B, and C is now Figure 1, and was moved to the Introduction section, following the Referee’s suggestion.
- It is not clearly explained in the Introduction, neither the main objectives of this study nor the expected outcomes, compared with previous information published by other groups (especially in reference 11).
Authors:
The Introduction was rewritten to make clear the objectives, outcomes and the Discussion compared with previous studies.
- The Introduction is not well structured because there is a description of the computational methodologies in the middle of the description of the protein-ligand interactions. The description of the covalent interactions between myricetin and 3CL protease is repeated (in lines 45-47 and 88-92).
Authors:
This part of computational methodologies was moved to the Materials and Methods section, and the text was corrected.
- The relevance of the conformational analysis carried out for the three ligands is not highlighted.
Authors:
The following paragraph was added: “The electronic structure of these complex molecules has a significant component from the conformational structure used. In this case, it is important to analyze the planarity of ring C and the torsion angle between rings B and C. Therefore, we have carried out a conformational analysis using CAM-B3LYP/def2TZV level and the implicit solvent integral equation formalism Polarizable Continuum Model (IEFPCM) method.”
- The MEP surfaces need a better explanation.
Authors:
The paragraph was corrected “The Molecular Electrostatic Potential (MEP) surface accounts for interactions in the molecule and is helpful in analyzing reactivity descriptor behaviors in non-covalent interactions. Figure 4 shows the MEP surfaces in kcal/mol of rutin, myricetin, and baicalein. The molecules show a different pattern regarding the charge distribution. Ring B has a negative distribution in rutin, and a positive distribution in baicalein, while myricetin is less homogeneous. Hydrogens of rutin in ring A have a positive moiety, and baicalein and myricetin are different due to the hydroxyls.”
- There is no reference for the results on myricetin (Table 1 and Table 2), and the reason for including them is not explained.
Authors:
Table 1 and 2 were calculated using CAM-B3LYP/def2TZV. In the literature, only the recent study of Farias and co-workers [DOI:10.1007/s00894-023-05468-w] has the values of quercetin for this same level of theory. However, for consistency, we have excluded the reference of quercetin in Tables 1 and 2.
- A plot of the best docking poses of the three ligands into the protein active site is needed.
Authors:
Figure 10 was added to analyze the best docking poses.
- The electronic structure calculations and the molecular descriptors are obtained for the three molecules in vacuum. The authors must explain why those results can be used to rationalize their ligand-protein interactions.
Authors:
The IEFPCM implicit solvent method was used and the results were also included. These descriptors including implicit solvent is important to study the ligands.
- “When the results of these studies are compared, it is possible to identify disagreements concerning the literature data [25], which the difference in the resolution of proteins used could justify. The 7JR3 protein used in the present study has better resolution (1.55 Å), which may relate to higher accurate results.” This statement must be clarified.
The paragraph was rewritten to clear this misprinting: “When the results of baicalein are compared, it is possible to identify one disagreement regarding the interaction with Cys145, which is with rings A and C while in the literature is with ring B…”
- The redocking results of myricetin should be better explained.
Authors:
This was a misprinting that was corrected, as there is no ligand in the PDB 7JR3.
- The quality of the Figures is low.
Authors:
The Figures have lowered the resolution when transferred to Word, now the problem was solved.
- English also needs revision.
Authors:
We have revised the English of the manuscript with the help of a native

Reviewer 2 Report
In this paper, molecular docking and electronic structure calculation were used to analyze three flavonoids. The paper is fascinating, and it is important to understand the mechanism of protease action. The article has the following problems that need to be revised:
1. On page 4, when talking about BCP which shown in Figure 2 and Table 1, radicals R1, R2, R5 and R6 were mentioned. These radicals should be marked in Figure2. In the same way, ring A, B should be marked in Figure3.
2. On Table 2, two “Myricetin” should be distinguished. ?LUMO was missed, the value of ???−? need to be checked, and the unit should be indicated.
3. On page 5, the discussion of HOMO-LUMO and nucleophiles-electrophiles is confused. The HOMO corresponds to the given electron. Rutin has a higher HOMO -159.74, can reactive as nucleophiles. The same problem arises in the discussion of Figure5.
4. On line 350, the author mentioned “The rutin presents four critical points with Hc < 0 and Hc/p< 0.” which conflicts with Table1. Only one critical points with Hc < 0 was shown in it.
5. The article still needs to be checked carefully.
Author Response
Reviewer #2
- On page 4, when talking about BCP which shown in Figure 2 and Table 1, radicals R1, R2, R5 and R6 were mentioned. These radicals should be marked in Figure2. In the same way, ring A, B should be marked in Figure3.
Authors:
We acknowledge the work done by the referee and the Figures of BCP and ELF were modified taking into account the referee suggestion.
- On Table 2, two “Myricetin” should be distinguished. ?LUMO was missed, the value of ???−? need to be checked, and the unit should be indicated.
Authors:
A column was included with the LUMO energies. The two entries of myricetin were clarified.
- On page 5, the discussion of HOMO-LUMO and nucleophiles-electrophiles is confused. The HOMO corresponds to the given electron. Rutin has a higher HOMO -159.74, can reactive as nucleophiles. The same problem arises in the discussion of Figure5.
Authors:
This confusion was due to the text that was not clear, there was a missing word to clarify that the center is a nucleophile and then react with an electrophile. We have corrected the text following the suggestion of the Referee.
- On line 350, the author mentioned “The rutin presents four critical points with Hc < 0 and Hc/p< 0.” which conflicts with Table1. Only one critical points with Hc < 0 was shown in it.
Authors:
The text was corrected.
- The article still needs to be checked carefully.
Authors:
We have deeply checked the text.

Reviewer 3 Report
The authors utilized a series of electronic structure analysis tools to investigate the conformational and electronic properties of rutin, myricetin, and baicalein, which are potential inhibitors to SARS-CoV-2 virus. In addition, protein docking is used to study their binding with 3CLpro protease.
The manuscript is publishable on IJMS after addressing the following questions.
1). The scientific question is not unclear. It should be made explicit what is the reason to study these inhibitors and their interaction types with 3CLpro residues provided the cited references such as Ref. 11 and 10.2174/1381612826999201116195851 in which molecular dynamics is also carried on.
2). The introduction should be rephrased, especially the second, third, and sixth, seventh paragraphs. For instance, the second paragraph explains the rutin-3CLpro binding while the third paragraph mentions the three inhibiters in a random order. They should be put in a more organized manner for each inhibitors and several interaction types such as covalent, hydrogen, nucleophilic, and electrophilic bondings. And move the two paragraphs of computational approaches to the end of the introduction.
3). The sentence in page 2 ``MEP calculations are also useful for understand the reactive behavior through van der Waals interactions (vdW).’’ is problematic. The vdW interactions are from dispersion, which differs from electrostatic interactions.
4). It is better to put the BCP analysis of covalent and non-covalent bonding together in subsection 2.2.
5). The MCP, ELF, and Fukui function analysis can prove the electrophilic and nucleophilic favorable attack regions, but how does the BCP analysis within the molecules demonstrate the ability of covalent and non-covalent bonding of the molecules with 3CLpro, which are inter-molecular interactions? On the other hand, the hydrogen-bonding is clear from the docking results in Tab. 3. This table indicates that the more H-acceptors and H-donors, the lower of the docking score of the inhibitors. Does it suggest that the hydrogen bonding dominate in the interactions?
6). The electronic structure analysis should also be carried on for quercetin for consistency, because it is added in the docking calculations.
7). The discussion in subsection 2.3 is not well ordered and confusing. The results of the molecules are better to present in the order as Fig 7. In addition, some discussions of the electronic structure results given after the docking in subsection 2.3 should be moved earlier to subsection 2.2, such as for the ELF, electronic density, and Fukui functions on page 10 because they are the conclusions that can be drawn from the quantum mechanics calculations.
8). The interaction type between Leu141 residue and quercetin from docking results is not clear on page 9. The sentence ``The Leu141 residue also establishes a clear interaction showing…’’ needs to be rephrased.
9). It is better to assign same colors to same residues in Fig. 7 to avoid confusion and put the residues at similar positions. In Fig. 7, the bindings of baicalein and myricetin look like reversed in the pocket. Why is the case? What if the molecules are binding at different directions? What the binding scores would be?
10). The docking results suggest that the interactions between pi-rings of the molecules and some residues are important. However, this type of interaction is missed in the electronic structure analysis. Or can it be included in the electron density analysis? Or the aromaticity can be reflected by the conformational results because the molecules favor planer configurations?
11). Could the authors propose several potential substitutes which might behave better in terms of binding score based on the interactions discussed in the manuscript?
Minor suggestions:
1). Move the figure 9 to the beginning and divide the flavonoids and sugar side chains by rectangular lines so that the structures and the referring of regions in the main text are easier to understand.
2). In the sentence in the introduction ``It was suggested that rutin interacts indirectly with protease through quercetin rings A and C and with residues His41 and Cys145…’’, one of the words ``and’’ and ``with’’ should be removed.
3). It is helpful if vertical lines are added into figure 1 to indicate the 0 or 180 degree.
4). HOMO and LUMO energies in Tab. 2 need units. And the third and fourth rows of this table have same names ``Myricetin’’, which might be a typo.
5). The colors green and purple in Fig. 5 need explanations in the title, ie, the Fukui functions are plotted by green and purple colors. And it is beneficial to label the ketone group and carbon 2, etc. out in this figure explicitly as in Fig 6. The label for baicalein ``(c)’’ seems to be a typo.
6). The abbreviation NCI needs to be defined first.
The sentences can be improved, especially in the result section. Many sentences are written as the facts but not the discussions drawn from the calculated data and results.
Author Response
Reviewer #3
1). The scientific question is not unclear. It should be made explicit what is the reason to study these inhibitors and their interaction types with 3CLpro residues provided the cited references such as Ref. 11 and 10.2174/1381612826999201116195851 in which molecular dynamics is also carried on.
Authors:
We acknowledge the work done by the referee. The aim of this work was included in the Introduction section. The main difference regarding the present work and those of the literature was that the recent study of Rehman and co-workers was based on molecular dynamics and docking of rutin (among other molecules) but not including baicalein and myricetin. Otherwise, the work of Su and co-workers have studied myricetin and baicalein including docking. The work of Cherrak et al. [https://doi.org/10.1371/journal.pone.0240653] studied rutin and myricetin using docking and molecular dynamics. Deetanya et al. [https://doi.org/10.1016/j.csbj.2021.05.053] studied rutin and baicalein using docking and a fragment molecular orbital method. Therefore, there was a question regarding rutin, myricetin and baicalein using the same electronic structure study and molecular docking protocols. The following text was inserted: “In the context of these molecules, 3CLpro is the target of relevance in the last few years [1–8,14]. However, studies of rutin, myricetin, and baicalein [14] have a gap in electronic structure descriptors, which are needed to evaluate new insights on the possible centers for the interactions. Most recently, 3CLpro [20] PDB 7JR3 with a resolution of 1.55 Å was determined [20]. Therefore, it is important to verify the binding modes of these molecules against the target 7JR3 [20].
This work aims to achieve important descriptors for designing new potential entities through electronic structure and molecular descriptors and studying the interactions through covalent, hydrogen bonding, nucleophilic, and electrophilic interactions. Moreover, 7JR3 PDB was used for the docking study. We have used Density Functional Theory (DFT) to obtain Fukui functions, bond critical points of Atom in Molecules Theory (AIM), frontier orbitals, non-covalent interactions, electron localization functions, and molecular electrostatic potential.”
2). The introduction should be rephrased, especially the second, third, and sixth, seventh paragraphs. For instance, the second paragraph explains the rutin-3CLpro binding while the third paragraph mentions the three inhibiters in a random order. They should be put in a more organized manner for each inhibitors and several interaction types such as covalent, hydrogen, nucleophilic, and electrophilic bondings. And move the two paragraphs of computational approaches to the end of the introduction.
Authors:
The Introduction section was completely rephrased following the referee suggestions.
3). The sentence in page 2 ``MEP calculations are also useful for understand the reactive behavior through van der Waals interactions (vdW).’’ is problematic. The vdW interactions are from dispersion, which differs from electrostatic interactions.
Authors:
The sentence was corrected.
4). It is better to put the BCP analysis of covalent and non-covalent bonding together in subsection 2.2.
Authors:
First, we have added the following: “Analysis of energy density ( ) and the electronic density distribution ( ) following the work of Cremer and Kraka [30][31] was used to study the covalent versus electrostatic character in a weak bond [32]. Bond Critical Point (BCP) (3,-1) can be used to investigate intramolecular hydrogen bonding, which is important for the interaction of the molecule with the target [13,32].”
Second, we have moved the analysis of BCP next to the paragraph discussing non-covalent interactions.
5). The MCP, ELF, and Fukui function analysis can prove the electrophilic and nucleophilic favorable attack regions, but how does the BCP analysis within the molecules demonstrate the ability of covalent and non-covalent bonding of the molecules with 3CLpro, which are inter-molecular interactions? On the other hand, the hydrogen-bonding is clear from the docking results in Tab. 3. This table indicates that the more H-acceptors and H-donors, the lower of the docking score of the inhibitors. Does it suggest that the hydrogen bonding dominate in the interactions?
Authors:
BCP analysis is important to investigate the hydrogen bonding of those ligands, and was discussed in the text of intramolecular descriptors. In order to separate these insights that are of intramolecular order from the rest of the text that are treating the ligand protein interaction. BCP was not properly introduced, and we revised and clarified this part.
6). The electronic structure analysis should also be carried on for quercetin for consistency, because it is added in the docking calculations.
Authors:
The electronic structure calculation was carried out in our recent study (https://doi.org/10.1007/s00894-023-05468-w), and this detail is now explained in the text. The citations of our recent study were included in the text, and now the Table is consistent.
7). The discussion in subsection 2.3 is not well ordered and confusing. The results of the molecules are better to present in the order as Fig 7. In addition, some discussions of the electronic structure results given after the docking in subsection 2.3 should be moved earlier to subsection 2.2, such as for the ELF, electronic density, and Fukui functions on page 10 because they are the conclusions that can be drawn from the quantum mechanics calculations.
Authors:
The electronic structure discussion on page 10 was moved to a new subsection 2.4. These discussions are the connections that were carried from the results of docking of subsection 2.3 against the information of subsection 2.2. In that way, the information from docking results is necessary to establish a link to ELF, MEP and Fukui Functions.
8). The interaction type between Leu141 residue and quercetin from docking results is not clear on page 9. The sentence ``The Leu141 residue also establishes a clear interaction showing…’’ needs to be rephrased.
Authors:
The text was corrected: “The Leu141 residue also establishes a strong short-range hydrogen bond interaction showing considerable closeness with quercetin and rutin ligands.”
9). It is better to assign same colors to same residues in Fig. 7 to avoid confusion and put the residues at similar positions. In Fig. 7, the bindings of baicalein and myricetin look like reversed in the pocket. Why is the case? What if the molecules are binding at different directions? What the binding scores would be?
Authors:
The colors are those used by the Discovery Studio program to identify and classify interactions between ligand and receptor. Baicalein and myricetin are in the same projection used in the literature: Ring A on the left, ring B on the right and ring C on the middle.
10). The docking results suggest that the interactions between pi-rings of the molecules and some residues are important. However, this type of interaction is missed in the electronic structure analysis. Or can it be included in the electron density analysis? Or the aromaticity can be reflected by the conformational results because the molecules favor planer configurations?
Authors:
The following was added: “This interaction reinforces the importance of aromaticity for the system to create effective interactions between the ligand and the catalytic dyad in the target protein.”
11). Could the authors propose several potential substitutes which might behave better in terms of binding score based on the interactions discussed in the manuscript?
Authors:
We are preparing a new manuscript regarding proposed new molecules based on the descriptors found in the present work.
Minor suggestions:
1). Move the figure 9 to the beginning and divide the flavonoids and sugar side chains by rectangular lines so that the structures and the referring of regions in the main text are easier to understand.
Authors:
Figure 9 was moved to the Introduction section and redrawn.
2). In the sentence in the introduction ``It was suggested that rutin interacts indirectly with protease through quercetin rings A and C and with residues His41 and Cys145…’’, one of the words ``and’’ and ``with’’ should be removed.
Authors:
The text was corrected.
3). It is helpful if vertical lines are added into figure 1 to indicate the 0 or 180 degree.
Authors:
Figure 1 was corrected.
4). HOMO and LUMO energies in Tab. 2 need units. And the third and fourth rows of this table have same names ``Myricetin’’, which might be a typo.
Authors:
Table 2 was corrected.
5). The colors green and purple in Fig. 5 need explanations in the title, ie, the Fukui functions are plotted by green and purple colors. And it is beneficial to label the ketone group and carbon 2, etc. out in this figure explicitly as in Fig 6. The label for baicalein ``(c)’’ seems to be a typo.
Authors:
The Figure and the caption were corrected.
6). The abbreviation NCI needs to be defined first.
Authors:
The text was corrected.
Comments on the Quality of English Language
The sentences can be improved, especially in the result section. Many sentences are written as the facts but not the discussions drawn from the calculated data and results.
Authors:
The results section was redrawn.

Round 2
Reviewer 1 Report
In this second version of the manuscript, the authors have answered most of my comments, but some paragraphs in the main text still need revision.
The sentence on page 13, "The results of the other flavonoids reinforce the hypothesis that higher degrees of freedom and higher hydrogen bond availableare related to the best score obtained by rutin." should be changed to "The results of the other flavonoids reinforce the hypothesis that a higher number of degrees of freedom and hydrogen bonds available are related to the best score obtained by rutin."
On page 14, the following paragraph should be rewritten: "Furthermore, when comparing MEP isosurfaces, it is possible to notice the intense electronic density adjacent to the non-conjugated ring of baicalein and rutin, which enables the interaction with the residue Cys145 (rutin) and His41 (myricetin). However, this interaction showed a weak nature and around 4.9 Å (Figure 9)."
The sentence on page 13, "The results of the other flavonoids reinforce the hypothesis that higher degrees of freedom and higher hydrogen bond availableare related to the best score obtained by rutin." should be changed to "The results of the other flavonoids reinforce the hypothesis that a higher number of degrees of freedom and hydrogen bonds available are related to the best score obtained by rutin."
Also, correct the following sentences: "It is possible to find in a first analysis that the rutin is the unique ligand capable of forming one hydrogen bond with the residue of the catalytic dyad His41/ Cys145. The best pose docking os these ligads shows the relevance of the active site."
Author Response
Reviewer #1
- The sentence on page 13, "The results of the other flavonoids reinforce the hypothesis that higher degrees of freedom and higher hydrogen bond availableare related to the best score obtained by rutin." should be changed to "The results of the other flavonoids reinforce the hypothesis that a higher number of degrees of freedom and hydrogen bonds available are related to the best score obtained by rutin."
Authors:
The sentence was corrected.
- On page 14, the following paragraph should be rewritten: "Furthermore, when comparing MEP isosurfaces, it is possible to notice the intense electronic density adjacent to the non-conjugated ring of baicalein and rutin, which enables the interaction with the residue Cys145 (rutin) and His41 (myricetin). However, this interaction showed a weak nature and around 4.9 Å (Figure 9)."
The sentence was corrected. “Furthermore, when comparing the MEP isosurfaces, it is possible to notice the intense electron density adjacent to the unconjugated ring, both in baicalein and rutin, which allows interaction with residues Cys145 (rutin) and His41 (myricetin). However, this interaction was weak in nature, with a distance of around 4.9 Å (Figure 9).”
- The sentence on page 13, "The results of the other flavonoids reinforce the hypothesis that higher degrees of freedom and higher hydrogen bond availableare related to the best score obtained by rutin." should be changed to "The results of the other flavonoids reinforce the hypothesis that a higher number of degrees of freedom and hydrogen bonds available are related to the best score obtained by rutin."
The sentence was corrected.
- Also, correct the following sentences: "It is possible to find in a first analysis that the rutin is the unique ligand capable of forming one hydrogen bond with the residue of the catalytic dyad His41/ Cys145. The best pose docking os these ligads shows the relevance of the active site."
The sentence was corrected. “In a first analysis, it is possible to verify that rutin is the only ligand capable of forming a hydrogen bond with the catalytic dyad residue His41/Cys145. The best docking pose of these ligands shows the relevance of the active site.”

Reviewer 3 Report
I have no further questions about the revised manuscript.
Author Response

(The authors gave the same response as above.)
